# Identifying the impact of COVID-19 on health systems and lessons for future emergency preparedness: A stakeholder analysis in Kenya

**Dosila Ogira** [1]*, **Ipchita Bharali** [2]*, **Joseph Onyango** [1], **Wenhui Mao** [2], **Kaci Kennedy McDade** [2], **Gilbert Kokwaro** [1], **Gavin Yamey** [2]

1 Institute of Healthcare Management, Strathmore Business School, Strathmore University, Nairobi, Kenya,
2 Center for Policy Impact in Global Health, Duke Global Health Institute, Duke University, Durham, North Carolina, United States of America

☯ These authors contributed equally to this work.
* dogira@strathmore.edu (DO); ipchita.bharali@duke.edu (IB)

**Data Availability Statement:** This paper has included all the data used in the analysis.

## Abstract

The coronavirus pandemic (COVID-19) has triggered a public health and economic crisis in high and low resource settings since the beginning of 2020. With the first case being discovered on 12th March 2020, Kenya has responded by using health and non-health strategies to mitigate the direct and indirect impact of the disease on its population. However, this has had positive and negative implications for the country's overall health system. This paper aimed to understand the pandemic's impact and develop lessons for future response by identifying the key challenges and opportunities Kenya faced during the pandemic. We conducted a qualitative study with 15 key informants, purposefully sampled for in-depth interviews from September 2020 to February 2021. We conducted direct content analysis of the transcripts to understand the stakeholder's views and perceptions of how COVID-19 has affected the Kenyan healthcare system. Most of the respondents noted that Kenya's initial response was relatively good, especially in controlling the pandemic with the resources it had at the time. This included relaying information to citizens, creating technical working groups and fostering multisectoral collaboration. However, concerns were raised regarding service disruption and impact on reproductive health, HIV, TB, and non-communicable diseases services; poor coordination between the national and county governments; shortage of personal protective equipment and testing kits; and strain of human resources for health. Effective pandemic preparedness for future response calls for improved investments across the health system building blocks, including; human resources for health, financing, infrastructure, information, leadership, service delivery and medical products and technologies. These strategies will help build resilient health systems and improve self-reliance, especially for countries transitioning from donor aid such as Kenya in the event of a pandemic.

**Funding:** This study is part of the ongoing project "Driving health progress during disease, demographic, domestic finance and donor transitions (the "4Ds"): policy analysis and engagement with six transitioning countries", under the project award No. OPP1199624, funded by The Bill and Melinda Gates Foundation. The funders had no role in study design, data collection and analysis, decision to publish, or preparation of the manuscript.

**Competing interests:** The authors have declared no competing interests.

## Introduction

Since the beginning of 2020, the COVID-19 pandemic has spread rapidly worldwide, causing devastating consequences for patients, health care workers, health systems, and economies [1]. As of January 14[th] 2022, more than 318 million cases had been confirmed and more than 5.5 million deaths were recorded worldwide [2]. Out of this, the African continent recorded 10,201,488 cases of COVID-19 and 232,770 deaths [3]. This represents approximately 3% of the total cases worldwide. In a bid to protect the population and mitigate the impact of COVID-19 infection, various efforts have made in research and development of vaccines, several of which have since been rolled out [4, 5]. With over 11.9 billion doses administered worldwide, 817 million doses were received in Africa out of which 577.8 million were issued as of May 2022 [2, 6].

The pandemic has put considerable strain on national health systems worldwide, including in relatively highly resourced settings [1, 7]. For instance, high-income countries (HICs), including those in Asia, Europe and North America recorded initial high morbidity and mortality rates [8]. This resulted in a surge in hospitalization rates, which saw the strain of healthcare workers and healthcare infrastructure, shortages in medication believed to alleviate COVID-19 symptoms and personal protective equipment (PPEs) [9–11].

The COVID-19 pandemic has also had profound consequences for resource-poor settings in low- and middle-income countries, including African countries [12]. The region is challenged by limited access to safe water and sanitation facilities, urban crowding and a large informal economy, creating added health risks [13]. Additionally, vulnerabilities in the health care system including, scarcity of resources such as oxygen and poor health infrastructure in the region create multiple health challenges in the era of the COVID-19 pandemic [14]. Many countries in Sub-Saharan Africa are also donor dependent. Large segments of their health systems are financed through external donors, leading to difficult trade-offs about interventions to prioritize [15–17]. In a bid to control infections, initial mitigation measures aimed at limiting the movement of people through lockdowns and quarantines were put in place. However, these policy directions also affected the access to other health services such as HIV, tuberculosis (TB), and malaria reversing the gains made in curbing these diseases [18].

## COVID-19 in Kenya

Kenya reported its first case of COVID-19 on 12[th] March 2020 [19]. Since then, the numbers have risen and as of January 16, 2022, almost two years after the confirmation of the first case, Kenya had recorded 317,634 cases and 5,488 deaths [20]. Like most countries globally, Kenya embarked on a countrywide vaccination program on March 5, 2021 [21]. As of January 16, 2022, the country had administered over 11 million vaccinations with the Country's capital Nairobi presenting the highest percentage uptake of 36% [20].

Throughout the COVID-19 pandemic, the Kenyan government has responded through various health and non-health strategies to mitigate the impact of the pandemic on its population [22, 23]. Some of the public health and socio-economic policies included; establishing the National Emergency Response Committee, international travel ban, closure of schools and workplaces, dawn to dusk curfew, provision of food aid, tax relief, and expansion of health insurance for healthcare workers [24–27]. Based on the Oxford Coronavirus Government Response Tracker, a composite measure of nine metrics calculating the stringency index of policy measures undertaken by countries to control the COVID-19 pandemic [28], Kenya's responses were considered moderate at the beginning. However, the measures progressively became high, peaking at 93.52 out of 100 from early May 2020 to late June 2020 with an increase in the number of cases [29].

Despite the government of Kenya putting the mitigation measures in place, concerns were raised over their effectiveness. For instance, due to the physical distancing measures, some groups were disproportionately affected including; those living in informal settlements, pregnant mothers, school going children, persons living with disability among others [30–34]. Additionally, cases of misappropriation of funds designated for COVID-19 were witnessed. This was believed to have a ripple effect, including crippling the country's ability to acquire sufficient medical supplies and employ an adequate number of human resources for health [35, 36].

Although major strides have since been made in fighting the COVID-19 pandemic with the development of vaccines, various lessons have been learnt for long-term health system strengthening to build resilience, including; global collaboration in crisis response, surveillance, stockpiling, health work force surge capacity, among other measures [37–39]. While diverse frameworks have been used to evaluate health system strengthening practices by countries, we adopted the Word Health Organization (WHO) health system building block framework to present the Kenyan case [40]. The framework outlines the interaction across the core components of the health system and has been widely applied for crisis response [41–43]. This study aimed to understand the key measures adopted in Kenya to tackle the COVID-19 pandemic, how the pandemic impacted the health sector and the population more broadly, and how future policy priorities and health emergency preparedness can be strengthened through the lessons learnt from the COVID-19 pandemic response.

## Methods

### Study setting and participants

We used purposive sampling to identify Kenya stakeholders from different national and county fields [44]. This included those who had a firm understanding of how the COVID-19 pandemic has affected the Kenyan healthcare system and were willing to take part in the study. A total of 15 virtual interviews were conducted with key informants, out of which, 5 represented government institutions (3 from the national level and 2 from the regional level), 3 represented donors and development partners, 5 representatives from healthcare professionals (3 providers and 2 from professional bodies) and 2 representatives from non-governmental organizations (NGO) and civil society organizations (CSO), respectively.

### Study design and data collection

The study used a qualitative cross-sectional design. We conducted 15 in-depth interviews using a semi-structured interview guide (S1 Appendix) which was developed to ensure that the desired area of inquiry was covered during individual interview sessions and to aid comparability of information obtained across the respondents. The interviews were conducted virtually between September 2020 and February 2021 by two research members of the team (JO and GK) with experience in conducting qualitative interviews. All supplementary notes were taken by one researcher (DO). All interviews were conducted in English and took an average of one hour. Interviews were recorded after obtaining oral consent from respondents.

### Data management and analysis

All interviews were transcribed and coded for analysis using NVivo software. Data analysis was done using the Framework approach [45]. Deductive content analysis was used for this study. We started with the WHO building blocks framework to guide the analysis and

further modified it based on findings to develop new themes not covered by the framework but mentioned in interviews. One researcher (DO) initially read through all the transcripts, line by line to develop an initial coding framework with input from (JO). This was then shared with (IB) who read through and double coded five transcripts, selected across the participant's category to refine the coding framework. With input from other study team members (GY, GK, WM, KKM), the differences in the coding framework were then reconciled and coding was done on the themes and sub-themes identified in the final framework. Two researchers (DO and IB) then applied the final coding framework to present results, which were then reviewed by the other study team members (GY, GK, JO, WM, KKM). The data was stored in a password protected shared directory on the Strathmore server based on Strathmore University ICT data protection policy. Additionally, since the study was collaborative research with Duke University, it approved data back up and sharing by Duke IT Security Offices. All personal identifiers were removed from the dataset prior to archiving in the Duke University data repository.

## Ethical approval and consideration

Ethical approval was obtained from Strathmore University's Institutional Review Board (0891/20) and the Duke University Campus Institutional Review Board (2019–0366). Informed consent was sought from participants send in advance through email. Since all the interviews were done virtually, verbal informed consent was obtained from all the participants after providing information about the study and the potential benefits and risks of their involvement. The interviews were conducted virtually to mitigate the risk to participants due to the COVID-19 pandemic.

## Results

The stakeholders discussed various dimensions of the COVID response in Kenya and identified key challenges and opportunities for future preparedness and response efforts. While taking into account the WHO's health systems building block framework [40], the findings are categorized into three broad themes: (1) Stakeholder perceptions on the country's COVID-19 response, which captures the views on the adequacy of resources used and the measures taken by the government to effectively in fighting the COVID-19 pandemic; (2) Impact of the pandemic on Kenya's health system and the population; (3) Opportunities to improve future pandemic preparedness and health system strengthening based on stakeholders' recommendations.

### Stakeholders' perceptions on Kenya's COVID-19 response

**Provision of emergency supplies.**   Majority of the respondents noted that the availability of supplies, such as personal protective equipment (PPEs), testing kits and reagents, was inadequate. This was attributed to the disruption of the international supply chain due to travel bans and border closures, which created a global shortage. Additionally, the high demand across countries resulted in an initial spike in prices of supplies rendering them expensive to acquire. However, while the country resorted to local manufacturing of PPEs to avert the shortages, concerns were raised about the poor quality and, therefore, the potential risk to healthcare workers exposed to infections.

"So, *in terms of the supply chain, we notice that the availability of PPEs was a problem, the cost was just out of this world, I think that at this point in time possibly we are buying PPEs at 10% or less of the cost that was reported in the beginning of the pandemic. And especially*

*most of the countries did not have access to COVID-19 test kits, so I would say our biggest challenge at that point was supplies.*"

*Key Informant 1*

Rigid procurement processes both at the national and county levels during the pandemic was faulted as one of the challenges that led to shortage of supplies. Respondents recommended that exceptions and favorable provisions should be made for emergencies.

**Surveillance and health information systems.**   Several concerns were raised regarding the robustness of the health information systems in the country at the beginning of the pandemic. It was noted that historically, the healthcare information system has been fragmented and most of the information, including patient records and files, are still paper based. This posed an initial challenge in receiving real-time information that could have been used for critical decision making, especially with the surge in COVID-19 cases. Few respondents pointed out that there is also an opportunity to embrace technology and digitize data that can be leveraged in critical decision-making.

"*Digital surveillance platforms are easier to analyze and could be producing all these dashboards in real time. I think we adopted it at some stage but in the earlier stages, I think we really were on paper-based approach which sometimes is hard to put on digital platforms and analyze and be able to make decisions.*"

*Key Informant 15*

Although there were efforts to undertake mass testing at the initial stage of the pandemic, the process was largely faulted by most of the respondents. First, the initial turnout was low, and the information received was not sufficient to make concrete policy recommendations. Secondly, it was noted that the tests were not being analyzed locally, causing delays and risks of transmission during the wait period. In terms of contact tracing, inaccurate contact information provided by some tested people presented a challenge in reaching them. One of the respondents attributed the provision of wrong information to the initial stigma associated with handling positive cases.

"*I think a lesson that we can learn, right from the initial stages, how do we approach contact tracing without necessarily coming closer to criminalizing it, I think that was the bigger challenge in the initial stages.*"

*Key Informant 15*

Almost all the respondents acknowledged that the country responded well in terms of sharing information on the COVID-19 pandemic with the public. They lauded the Ministry of Health for continually informing the public on developments regarding the pandemic. Some of the respondents felt that cross-border exchange of information from countries that were already experiencing the pandemic, such as China, provided an opportunity for Kenya to put stronger mechanisms in place and improve preparedness.

"*. . .I must commend the government and the Ministry of Health in terms of giving information to the public. We had enough materials circulated in the media and even through the facilities. We had regular memos from the Ministry of Health and particularly the acting director-general informing the healthcare workers in terms of what needs to be done.*"

*Key Informant 3*

**Availability of human resources for health.**    Several challenges were highlighted regarding human resources for health during the pandemic. First, respondents pointed out that the available workforce was inadequate and misallocated and poorly trained on the management of COVID-19. While there was a bid to increase the workforce through temporary hiring and redeployment from other programmes to COVID isolation and quarantine sites, some of the respondents felt that this move was not well thought out since: (i) the hiring was done on a short-term basis and posed a challenge to sustainability in the long run; (ii) the few workers left in the facilities were stretched and not working efficiently. Secondly, the inadequate supply and poor quality of PPEs created fear of infection among the healthcare workers, taking a toll on their mental health due to concerns about exposing themselves and their families to the COVID-19 infection.

"*The number of the healthcare workers that were available, number two the protection of healthcare workers by offering quality protective PPEs, and number three in terms of training. You realize from the Ministry of Health data, most counties were below per in terms of the number of people that were training for COVID-19, and you notice in some areas we had some health workers running away when they heard patients had signs of COVID and this shows anxiety among them because of lack of training...*"

**Key Informant 1**

**Adequacy of health infrastructure.**    There was consensus most respondents regarding the inadequacy of healthcare infrastructure. Some of the respondents' challenges were the government's capacity to provide adequate quarantine facilities, leading to overcrowding in the few designated and posing a more considerable infection risk. Additionally, the pandemic revealed the initial insurmountable capacity gap of Intensive Care Units (ICU), with approximately 500 ICU beds available across the entire country to care for critical patients. As a result, some patients lost their lives due to lack of hospital bed space for critical care services.

"*.. the public health system is not well equipped in terms of the facilities, in terms of the equipment...The challenge which the Kenyan health system has faced mainly is number one capacity to accommodate those people requiring admission... We don't have capacity in terms of hospital beds, in terms of ICU capacity and then the number of facilities we have are very limited.*"

*Key Informant 3*

**Adequacy of financing for COVID-19.**    Most of the respondents felt that the health sector in Kenya is significantly underfunded and was further strained by the COVID-19 pandemic. Some argued that the onset of the COVID-19 pandemic created competing needs in the healthcare sector, thereby necessitating efficient and effective way of prioritizing and coordinating the financial resources.

"*The outright answer is our resources have not been enough; both financial, supplies and by a large extent... If you look at the budget allocation in the health sector, we have been oscillating between 5.6% and 6% or about 6.7% over the last 4 to 5 years, against the Abuja Declaration of 15%. If you look at it from the GDP point of view, we have to push for about 5% of GDP going into the health sector, I think we are oscillating between 1.5% and 2%, which means we are still way below the financing and therefore if anything comes on board that destabilizes the balance...*"

*Key Informant 14*

A few of the respondents raised concerns regarding the misuse of funds that had been mobilized domestically and from donors to curb the pandemic. Coordination of funding priorities between national and county levels was also highlighted as a challenge. Other issues including improper utilization of funds, delayed disbursements, skewed priorities and lack of expertise among officials were said to impact health financing decision-making.

"*I think for COVID, and we don't know how many other pandemics we are yet to get into, is how efficient we are in our Public Finance Management, especially in fund flows to getting the money to where it is needed in good time. . . .we failed in terms of timely disbursements. This serves a lesson for in future how do we get such emergencies taken care of in good time, to get the money where it is needed.*"

*Key Informant 15*

When asked about the role of external aid and support in facilitating the COVID response in Kenya, respondents talked about instances where various local and international actors, and agencies offered financial and technical support to the country, including supplies such as PPEs and testing kits. Some agencies, such as USAID were said to have repurposed some of the funds to optimize the fight against the pandemic, while others such as the World Bank offered technical and financial support.

"*From the World Bank, we have these multilateral agreements. . . one of them was activated very quickly to make that 5 billion Kenya Shillings (Approximately 50 million USD) available. There was support from the EU for example, there was support from DANIDA, and then there was some support from the US government also through USAID and others but working through their implementing partners. And of course the local contribution from the private sector through the resource mobilization committee.*"

*Key Informant 8*

**Coordination between the national and county governments.** Despite the initial move to set up an inter-governmental and multi-sectoral emergency response committee comprising of the health, security, education, transport, finance and trade sectors, some respondents felt that it was poorly executed in the beginning, with unclear roles and each arm operating autonomously in a situation that called for collaboration. Concerns were also raised by some of the respondents regarding the coordination between the national and county governments. For instance, the COVID-19 isolation centers were initially set up at three hospitals in the country's capital. Patients who lacked alternatives in their own counties were turned away due to poor referral systems and overcrowding.

"*When surveillance was devolved, contact tracing and all these things, we saw that hampered very much by the ability of the county to activate or facilitate response teams. When the county failed, they said the county failed and yet this is a national emergency.*"

*Key Informant 12*

However, opportunities were also leveraged through enhanced county level responses as illustrated by one of the respondents;

"*. . .and borrowing the lessons of COVID-19, we must remain alert, prepared and be able to work together. And I can give you an example. In my county, my governor set up different committees and I chair one committee where all development partners with a county commissioner and the governor himself, we all sit down to track how the pandemic is moving and mobilize additional resources.*"

*Key Informant 5*

## Impact of COVID-19 on the health sector, population and the economy

**Impact on health services provision.**   The onset of the COVID-19 pandemic in Kenya presented a shift in the provision of some healthcare services deemed non-essential. This saw the government closing some of the outpatient clinics and peripheral facilities and reallocating resources, including human and financial, to cater to the COVID-19 response. Majority of the respondents cited that health services, including maternal and child health (MCH), non-communicable diseases (NCDs), HIV, TB and elective surgeries were negatively affected. Under MCH services, sexual reproductive health, family planning services and immunization had to be stopped periodically. Additionally, it was pointed out that some counties converted their maternal units to COVID-19 isolation units, which impacted mothers' access to care. Although various policies were developed and put in place, some of them lacked clarity, including those for essential and emergency services, hence negatively affecting service provision and health seeking behaviours among the public. Fear of contracting COVID-19 and seeking services past curfew hours as well as capacity and supplies gaps were also highlighted as some of the reasons as to why most people avoided seeking care at health facilities, with others resorting to home based care.

"*You find that antenatal care is considered elective therefore, mothers did not go, even immunization was considered elective, therefore, children did not go for immunization, so those services were affected. And also, family planning access may have been seen as elective and further on surgery, elective surgeries, NCDs, checkups and clinics, medical clinics and surgical clinics may have been considered elective.*"

*Key Informant 12*

Some of the respondents also reported that the pandemic highly impacted patients, especially those who needed continuous and routine care (i.e., cancer patients seeking care in the country's capital Nairobi), due to the imposed lockdown and cessation of the provision of these services that were now considered elective.

"*Many people, including cancer patients who used to come and get their chemotherapy, and get their radiotherapy, those services went down dramatically. . .some people who were waiting to be given chemotherapy, cancer patients, could have missed several cycles and perhaps lost their lives.*"

*Key Informant 3*

**Impact on the pathway towards UHC.**   While discussing the long-term impacts of the COVID-19 pandemic on the country's journey towards achieving Universal Health Coverage (UHC), respondents felt that progressively, this would result in decline in coverage and reverse the gains that have been made in these fronts in the country, especially in the case of NCDs and routine care services, like cancer treatments, dialysis etc. Respondents also pointed out that the government was not providing health insurance for the larger population, including

the healthcare workers at the onset of the pandemic. Individuals were expected to make out of pocket payments which led to instances of financial hardship. This also affected the willingness of individuals to come forth to get tested or seek treatment at designated COVID-19 facilities.

" *One of the big impact of COVID is the fact that some of the other health conditions, have fallen back behind and therefore it means that in our attainment of UHC there is a lot more that will need to be done because now. I am sure we will have more people affected by different conditions and most notably I would say the NCDs, one, either because people have not then been seeking care at the health facility because of the perception that they will actually get infected, and maybe not taken their medication in the right way that they should.*"

*Key Informant 13*

**Impact on vulnerable population groups.**    Almost all respondents acknowledged that although the COVID-19 pandemic affected the whole population, there were certain sub-sets that were more negatively impacted. One of groups singled out by majority of the key informants were those working in the informal sector or daily wage earners, who constitute almost 80% of the Kenyan population and mostly reside in informal settlements. This is because some of the initial containment measures put in place including lockdown and closing some sectors of the economy such as bars and restaurants, increased their vulnerability by affecting their jobs and livelihoods. Additionally, public health measures that required the purchase of masks and sanitizers presented a challenge to those living in informal settlements and with limited resources.

"*. . . we are aware that with that [COVID-19 pandemic] came quite a number of restrictions that of course closed the economy and we know that over 80% of Kenyans are either poor or near poor, meaning they are one incident away from poverty, so any single incident will push them into poverty and COVID-19 is one of such incidences where if they don't get a salary for one month then they would be literally be below the poverty line.*"

*Key Informant 14*

Some stakeholders pointed out that women and girls were disproportionately affected by the pandemic. Cases of gender-based violence, especially against women, were on the rise due to economic stress in households and social isolation resulting from movement restrictions. Additionally, some stakeholders pointed out that the school system offered security to girls from communities that practice early marriages and female genital mutilation, and school closure resulted in an increase in these cases.

"*On one side, communities that practice early child marriage and female genital mutilation, we saw these things increasing because now girls were at home, they were more vulnerable, they were not going to school. . .girls who come from poor families and rural communities who would depend on the government supply of sanitary commodities could no longer access them because now they were at home and those sanitary commodities are largely supplied through schools.*"

*Key Informant 12*

Commenting on the pandemic's impact on school and education, a few of the respondents mentioned that children in rural and remote settings were affected by the temporarily closure of learning institutions due to lack of access to the internet and laptops. Additionally, some of

the children in the rural counties relying on government-supported school feeding pro-
grammes saw reduced access to food. Few respondents noted that children with special needs,
who mostly rely on teachers with special needs training, affected their learning. Additionally,
people with disability were also affected due to the social distancing measures put in place.

"*. . . and then you have persons who are disabled so they need physical support, they actually
need someone to pull and to push their wheelchair, or they need someone to hold them and
help them get into a matatu [minibus used for transport], and so on. . .so, this physical dis-
tancing measures were disproportionately affecting people that are blind, people that cannot
walk, people that cannot talk; so, the disabled were disproportionately affected.*

*Key Informant 4*

**Impact on donor transitions in the health sector.**   Due to the heavy reliance on donor
funding in the health sector, majority of the respondents expressed their concerns about its
impact on the health sector more generally, and the impact of COVID-19 on donor transitions
in Kenya. Some of them felt the donor countries are likely to shift their resources to focus
more on their own needs in dealing with the pandemic. In contrast, others felt that donors
would reevaluate transition timelines and be more forthcoming to boost investments in a bid
to curb the pandemic and strengthen health systems.

"*. . . most countries having experienced the pandemic and economic crisis which they have not
had in the past, we expect that they will focus more on their individual country's needs as
opposed to donations, and of course lower middle-income countries like Kenya, we need to
prepare for that and set priorities in the health system to ensure that the little funds that we
have are used in an efficient way.*"

*Key Informant 2*

## Opportunities to improve future pandemic preparedness

**Greater financial flexibility and improved coordination to respond to pandemics.**   Var-
ious recommendations were made to improve health financing arrangements and strengthen
financial prioritization and coordination to tackle future health emergencies. First, there were
suggestions to create an emergency fund within the Ministry of Health that can be tapped and
easily accessed in emergencies. Second, stakeholders urged for reforming the public financial
management laws to allow flexibility and improve financial decision-making during an emer-
gency. Third, respondents called for introducing financial laws and regulations that are
responsive to unique situations such as pandemics that would facilitate improved fund utiliza-
tion at the national and county levels. Respondents argued that counties and facilities should
be given the financial autonomy to carry their duties, such as hiring more health workers dur-
ing an emergency without overtly relying on the national government. Respondents also called
for fostering stronger public private partnerships to mobilize resources to tackle future
pandemics.

"*For financing, we must have an emergency fund that is backed by law, that this percentage
must be put for emergencies even though it keeps revolving every year. Because, if we have to
start forming committees to get funds or to start fundraising now, you see the delays in the
response.*"

*Key Informant 9*

**Improving self-reliance through increased domestic health investments.** Respondents generally agreed that Kenya should prioritize resource mobilization and spend efficiently to minimize the financial strain and service gaps resulting from the COVID-19 pandemic and impending donor transitions. Respondents urged for better donor transition planning and improved accountability in using available external resources to build a resilient health system. Apart from improved resource mobilization, few respondents cited that the country should emphasize efficiency improvements in the health sector by adopting mechanisms such as health technology assessments. Additionally, there were views to foster a more robust consultation between African countries, the national and county governments in resource allocation, and leveraging on public-private partnerships to seal the gap that will result from donor exit.

"*I would ask that especially in the health sector, we adopt health technology assessment as a key intervention that helps us understand where we have the highest return on investment. We do not need to add more resources maybe right now, but we need to ensure that we know where our money is and what our money is doing and looking at how best can we maximize on our efficiencies.*"

*Key Informant 14*

" *25% of the Kenyan healthcare sector is financed by donors. . .we have transitioned into a lower middle-income country. . . when you look at HIV AIDS, vaccines, malaria in the country, the dependency is <u>much</u> higher. If we don't have a plan for how we will replace the funds that we get from donors, then we are going to lose the gains that we have made on those specific disease. . .*"

*Key Informant 4*

**Improved financial protection for individuals to achieve UHC.** Given the catastrophic health expenses borne by families at the onset of the pandemic due to lack of coverage by both public and private insurance schemes, there were suggestions to increase protection through social and private insurance that can be adjusted to accommodate the larger population in instances of a pandemic. Additionally, to increase the country's health system resilience, there were suggestions to increase equity in resource allocation, coupled with political goodwill in a bid to achieve UHC.

"*In terms of health financing and UHC, COVID presents a fantastic opportunity for us to reengineer our health systems;. . .no one is safe until everyone is safe. . .if we don't bring everyone under a mechanism of ensuring that they have access to care then it does no good to all of us because the fact that your neighbor is not covered or is unable to access a treatment on COVID or preventive measures on COVID, then that means you are not protected in the first place.*"

*Key Informant 15*

**Addressing gaps in health infrastructure.** Despite the challenges linked to infrastructural gaps, some of the respondents reported that setting up urgent health facilities created an opportunity for increased structural capacity that can still be used post-COVID. There was

also recommendation to increase investments in health systems infrastructure such as ICU to cater to future pandemics and other ailments.

"*Now, we have been able to put capacity in most of our health facilities, there are counties which would not have had ICUs in many years to come. I am sure even after COVID, those ICUs will be used for other ailments going forward.*"

*Key Informant 2*

**Strengthening human resources health.** There were suggestions to rethink human resources development sustainably, including expanding the health workforce and greater focus on tackling health emergencies. Additionally, some respondents also highlighted the need to continuously train the healthcare workforce on emergency preparedness by embedding it in their curriculum to create better and timely response in case of future pandemics.

"*The workforce, the preparedness among our people to deal with the pandemic needs to be done <u>well</u> in advance. . . I mean, we know this might happen. It should becomes part of our curriculum in our medical schools and nursing schools and schools of public health. . .*"

*Key Informant 6*

**Fostering cross-sectoral collaborations for maintaining essential health services during health emergencies.** Several respondents mentioned that despite these challenges, the pandemic provided an opportunity for multisectoral collaboration, which helped ease the pandemic's impact. In terms of progressive response, respondents mentioned that, through partnerships fostered between the Ministry of Health and private sector players, guidelines and outreach for MCH, TB and HIV programs filled the initial service provision gaps created by the pandemic. There were recommendations for the government to invest in the delivery of essential services during a pandemic in two major ways; prioritization of continuity of services and dedication of funds for the provision of essential services.

"*.. we have to put a lot of effort towards maintaining the essential healthcare services that have been going on. Indeed services were negatively affected, not that people stopped being sick, but people feared the pandemic, they did not seek healthcare services. So, even as we respond to any pandemic, we also need to be aware that we need to respond to the existing conditions. . .*"

*Key Informant 10*

Various opportunities were witnessed in the country's policy response and measures, including creating local testing capacity by leveraging technology and innovation and strengthening public and private sector collaborations. Additionally, some of the respondents recommended that there should be a deliberate effort to map and support the vulnerable population since their economic and social state directly correlates with the larger health outcomes.

"*Our second level of preparedness should now be looking at the impact of each sector of the economy and mobilize those multi sectoral responses for mitigation. Those mitigation measures in each of those sectors of the economy are what is going to put in place a firm foundation for dealing with potential long-term impacts, making sure that there are certain policy changes that may have to be addressed.*"

*Key Informant 8*

Stakeholders also noted that there is an opportunity to strengthen local manufacturing of healthcare commodities to help reduce import costs and mitigate shortages in instances where the global supply chain is affected.

"*We have also realized that in the very initial stages we were importing some of the very simple materials from China and other countries. As we talk now, in the country, actually we deliberately decided to take a route where we are creating capacity to be able to produce things locally. The net effect is that there was serious significant reduction of cost in terms of what we were spending to access some of those things*!"

*Key Informant 2*

**Incorporating lessons learnt from previous health emergencies and other country experiences.**   The majority of the respondents acknowledged that lessons learnt from previous pandemics, including SARS, Ebola, and HIV were progressively incorporated and helped Kenya leverage the existing systems and policies to fight the COVID-19 pandemic.

"*But as it were, before the pandemic, we had the laws and the policies which were actually supposed to direct us on what to do in case of any new emerging disease. And they are very many, as you can recall, we had SARS, we had Ebola. . . Although the initial reaction was not immediate and some services had been disrupted, we quickly adopted some of the strategies we had.*"

**Key Informant 9**

Most of the respondents felt that the capacity of all health systems across the world was tested during the pandemic. However, some mentioned that countries in the continent such as Senegal, the Democratic Republic of Congo, and Uganda had dealt with previous pandemics such as Ebola in earlier years and had better surveillance and response to the pandemic since they leveraged existing systems. Respondents suggested that Kenya should use a blend of lessons from all the countries to develop a solution that will work best in the Kenyan context.

"*If you talk around institutionalizing disease surveillance as a long-term thing, I think Uganda is a country we can learn from. They have a reasonably good disease surveillance process, virology centre they have built, and I think this is because of their Marburg and Ebola outbreaks in the past. Senegal started off with a very early lab information system where all tests were put onto a lab information system, you could see who is testing, how the tests are followed. So, we may not learn the whole response from one country, but we can learn aspects country by country.*"

**Key Informant 12**

## Discussion

This study explored perspectives of key stakeholders in Kenya's health system on the country's response to the COVID-19 pandemic, its impact on the health sector, and implications for future pandemic preparedness. The COVID-19 pandemic presented Kenya with multiple challenges that disrupted the health system and had ripple effects on the entire economy. Shortcomings related to the WHO building blocks were mentioned frequently in our study. Given their complex interaction, there is need to strengthen the healthcare system in the event of a

future pandemic of similar magnitude. Addressing these challenges can contribute to improved responsiveness, risk protection of the population and delivery of quality and efficient health services.

The initial negative impact of the pandemic on continuity of essential and non-essential/ general health services was revealed in our study. MCH, TB, HIV, assistance for patients requiring routine and continued care were some of the categories highlighted, with attribution to factors such as government directives to discontinue these services and redeployment of staff to offer COVID-19 relief and support services being mentioned. These findings were comparable with other studies from both high and low resource settings which saw a significant disruption in health service provision [31, 39, 46–50]. However, not all services were affected as outlined in these studies from Kenya [51] and Ethiopia [52]. As a COVID-19 post recovery strategy for protecting the public health gains made for these services, it is pertinent to strengthen governance, coordination and informed decision making across the health service delivery network [53–55]. This will help in promoting provision of quality health services that are essential in ensuring achievement of UHC even in times of public health crisis [55].

Reduction in individuals' health-seeking behavior, as seen in our study, was also seen in other countries [56, 57]. This was associated with the discontinuation of some health services, lockdowns, curfews, and the fear of infections. Previous studies undertaken in past pandemics such as Ebola have also outlined changes in health seeking behaviors [58, 59]. As recommended by our key informants and seen in other studies, a key measure to ensuring continuity of services in the event of a pandemic is to foster multi-sectorial collaboration and developing a resilient health system that is able to cater for needs from a pandemic while maintaining routine health services [41, 60, 61].

Apart from the gaps in effective pandemic response, our study highlights several existing issues such as inadequate healthcare financing, lack of infrastructure, and human resource capacity constraints that have important implications for achieving UHC in Kenya. These challenges can severely impact overall health system resilience, especially as Kenya is undergoing a transition from concessional donor assistance and needs to become more self-reliant in providing services for its population [62]. Strengthening primary health care and adopting the right mix of Global Health Security (GHS) and UHC domains has been argued as an approach to resolve the health system gaps [63, 64]. Additionally, improving public financial management for improved budget allocation and accountability can be integrated by the Country to enhance its future preparedness [65, 66].

The respondents in our study revealed the authorities' initial shortcomings to manage adequate provision for testing, isolation, and quarantine services. However, these services are seen at the heart of effective public health responses to COVID-19. Respondents noted that while the government took appropriate public health measures to curb the pandemic, it fell short on several fronts owing to overwhelmed health facilities and personnel, lack of adequate resources, and issues with capacity and coordination. These findings mirror those from an analysis to understand lessons that Kenya can learn from the pandemic while linking it to historical gaps in the country's health system [67]. The resource shortages were attributed to long-standing gaps in the health sector stemming from poor leadership and governance that have preceded the pandemic. Good leadership and governance form part of the key ingredients that determine how a country respond's to public health emergencies [68].

Our findings highlighted the prompt response of the government to introduce measures to curb the spread of the virus, share information and raise awareness about the pandemic among the public. These findings are similar to that of two other studies aimed at assessing the knowledge, attitude and practices among the youth and households in informal settlements in Kenya [69, 70], and a scoping review assessing the same for Sub-Saharan Africa [71]. In the

three studies, the results indicated that there was high knowledge of the COVID-19 symptoms and preventive strategies, owing to active awareness campaigns by various governments. However, these studies also revealed that a high level of knowledge does not necessarily translate to preventive measures. Respondents raised concerns about the robustness of the HIS and mentioned that its fragmentation hindered timely relay of information for critical decision making at the onset of the pandemic. The use of information technology has been seen to act as an enabling factor for health care utilization by increasing the availability and accessibility of health services, especially for people from rural and remote areas, which will also make it more affordable and less time constraints [72–74]. Additionally, adopting a framework that collects the right data that can be used for future forecast in the event of a pandemic, both at the national and subnational level, is essential in strengthening the country's health security [75, 76].

## Strengths and limitations

This study focused on Kenya's health system's early responses and overall preparedness to tackle COVID-19. The interviews were conducted during the second wave of the COVID pandemic in Kenya while response measures were still evolving. While the study focuses on various public health measures and controls adopted to curb spread of the pandemic, it does not focus on COVID-19 vaccination strategies, which is critical to ending the pandemic. Participants in the study do not include health care service recipients during the pandemic.

This is among the first studies in Kenya that focused on understanding the impact, response, and policy implications of the COVID-19 pandemic in Kenya through interviews with key stakeholders who were involved first-hand in handling the pandemic in Kenya. The study results provide important insights for future policy and planning to respond more effectively and deliver health services during future health emergencies in Kenya.

## Conclusion

This study provides an overview of the early responses to the COVID pandemic in Kenya, pointing out the impact and key challenges that affect how Kenya can improve preparedness for dealing with future pandemics. Improving health sector investments by identifying strategies to minimize the effects of essential health systems could help improve pandemic response in the future. Stakeholders also called for better coordination, more flexibility in financial decision-making, and improved self-reliance to manage the pandemic better.

## Supporting information

**S1 Appendix. Topic guide for key informant interviews.**
(DOCX)

## Acknowledgments

We would like to acknowledge Dr. Indermit Singh Gill, current Vice President for Equitable Growth, Finance and Institutions at the World Bank and former Professor of Public Policy at Duke University and Dr. Osondu Ogbuoji of Duke University, Center for Policy Impact in Global Health for providing valuable insights and guidance in designing this study. The authors also wish to thank all key informants who provided valuable information for the study.

## Author Contributions

**Conceptualization:** Ipchita Bharali, Gilbert Kokwaro, Gavin Yamey.

**Data curation:** Dosila Ogira, Ipchita Bharali.

**Formal analysis:** Dosila Ogira, Ipchita Bharali, Joseph Onyango.

**Funding acquisition:** Gavin Yamey.

**Investigation:** Dosila Ogira, Joseph Onyango, Gilbert Kokwaro.

**Methodology:** Dosila Ogira, Ipchita Bharali, Joseph Onyango, Wenhui Mao, Gilbert Kokwaro, Gavin Yamey.

**Project administration:** Dosila Ogira, Ipchita Bharali, Kaci Kennedy McDade, Gilbert Kokwaro.

**Supervision:** Joseph Onyango, Wenhui Mao, Gilbert Kokwaro, Gavin Yamey.

**Validation:** Ipchita Bharali, Joseph Onyango, Wenhui Mao, Kaci Kennedy McDade.

**Writing – original draft:** Dosila Ogira, Ipchita Bharali.

**Writing – review & editing:** Dosila Ogira, Ipchita Bharali, Joseph Onyango, Wenhui Mao, Kaci Kennedy McDade, Gilbert Kokwaro, Gavin Yamey.

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
