## [Decision Letter · Decision Letter 0]

10 May 2022

PGPH-D-22-00331

Identifying the impact of COVID-19 on health systems and lessons for future emergency preparedness: a stakeholder analysis in Kenya

Dear Dr. Ogira,

Thank you for submitting your manuscript to PLOS Global Public Health. After careful consideration, we feel that it has merit but does not fully meet PLOS Global Public Health’s publication criteria as it currently stands. Therefore, we invite you to submit a revised version of the manuscript that addresses the points raised during the review process.

Please submit your revised manuscript by . If you will need more time than this to complete your revisions, please reply to this message or contact the journal office at globalpubhealth@plos.org. Please include the following items when submitting your revised manuscript:

We look forward to receiving your revised manuscript.

Kind regards,

Veena Sriram

Academic Editor

Journal Requirements:

1. Please provide a/amend your detailed Financial Disclosure statement. This is published with the article. It must therefore be completed in full sentences and contain the exact wording you wish to be published.

- Please clarify all sources of funding (financial or material support) for your study. List the grants (with grant number) or organizations (with url) that supported your study, including funding received from your institution. 

- State the initials, alongside each funding source, of each author to receive each grant.

Please ensure that Funding Information matches with the Financial Disclosure Statement.

2. Please provide an Author Summary. This should appear in your manuscript between the Abstract (if applicable) and the Introduction, and should be 150–200 words long. The aim should be to make your findings accessible to a wide audience that includes both scientists and non-scientists. Sample summaries can be found on our website under Submission Guidelines: 

https://journals.plos.org/globalpublichealth/s/submission-guidelines#loc-parts-of-a-submission

Additional Editor Comments (if provided):

Reviewers' comments:

Reviewer's Responses to Questions

**Comments to the Author**

1. Does this manuscript meet PLOS Global Public Health’s publication criteria? Is the manuscript technically sound, and do the data support the conclusions? The manuscript must describe methodologically and ethically rigorous research with conclusions that are appropriately drawn based on the data presented.

Reviewer #1: Yes

Reviewer #2: Yes

2. Has the statistical analysis been performed appropriately and rigorously?

Reviewer #1: N/A

Reviewer #2: N/A

3. Have the authors made all data underlying the findings in their manuscript fully available (please refer to the Data Availability Statement at the start of the manuscript PDF file)?

Reviewer #1: Yes

Reviewer #2: No

4. Is the manuscript presented in an intelligible fashion and written in standard English?

Reviewer #1: Yes

Reviewer #2: Yes

5. Review Comments to the Author

Reviewer #1: Line 104: The author's objective was to understand how the pandemic impacted the health sector and the population more broadly. Consider revising the "impact on population" and focus on the impact on the health sector since population data was not covered in the manuscript.

Line 109 - Please share some information on how you purposively sampled. Add inclusion criteria for purposively sampling the KIIs

Line 121 to 133 - The roles of JO, DO, GK and IB. Add a section for the author's contributions. i.e. GY, KKM, WM roles are not clear

Reviewer #2: Major comments:

Overall, the paper addresses a timely topic (that of offering evidence to strengthen health systems and ensure their resilience in future pandemics) while contextualizing this in the Kenyan context. The methods itself appear to meaningfully contribute toward this goal, and writing is relatively clear.

However, the paper largely seems to be lacking in effective use of the literature on this topic. For example, the authors in the abstract suggest links needed with the 6 WHO health system building blocks, but never refer to this framework in the paper or engage with the literature on how these approaches can help to improve pandemic preparedness. Similarly, the paper attempts to draw a link between health systems for health security and UHC, but does not draw on recent research on this subject (e.g., Lal et al., Erondu et al., Assefa et al., etc.) -- this is necessary to help further the arguments and structure of the paper. Finally, it may be helpful to define key terms, like UHC (the paper uses this term more as an approach for universal health care, not addressing financial burden as UHC is technically defined) as well as "resilience" of health systems.

Minor comments:

- Intro paragraph notes "9 million doses" worldwide -- seems like a typo and should be "billion"? Also, would be good here to note how many went to African continent, since intro makes distinction between global cases and Africa cases of COVID-19.

- Para 2 -- worldwide is mistakenly repeated after specifically referring to European and American countries ("American" countries is also confusing).

- In line 91 -- what are "high" measures?

- In the section starting at line 420, title "Opportunities to improve future pandemic preparedness and UHC" -- where does the UHC part come in? This seems to mostly focus on pandemic preparedness. Meanwhile, there is a section soon after on financial protection -- perhaps here would be better to elaborate on UHC.

- In paragraphs starting at line 480, why is infrastructure and human resources considered together, when in previous (1) section, these are separate sub-headings?

- In paragraph starting at line 535, there is a disparity between "majority of the respondents acknowledged that lessons learnt from previous pandemics, including SARS, Ebola, and HIV, helped Kenya leverage the existing systems and policies to fight the COVID-19 pandemic" + "A key lesson learnt for continuity of essential services was to continue having outreaches" vs. your earlier depiction that previous lessons on this were not applied and essential services were severely disrupted during COVID-19 as well -- would be helpful to offer an explanation of why lessons were not learned, otherwise the argument of using lessons learned appears to fall apart.

- The discussion section could benefit from more rigorous analysis and reflection on the findings. As presented, it reads mostly like a summary of the results section, except for a few references linking to similar studies. More meaningful incorporation of other academic research, relevant evidence from other contexts, and implications for what this truly means in terms of specific recommendations would significantly improve the paper.

6. PLOS authors have the option to publish the peer review history of their article (what does this mean?). If published, this will include your full peer review and any attached files.

**Do you want your identity to be public for this peer review?** For information about this choice, including consent withdrawal, please see our Privacy Policy.

Reviewer #1: No

Reviewer #2: No

---

## [Decision Letter · Decision Letter 1]

19 Sep 2022

PGPH-D-22-00331R1

Identifying the impact of COVID-19 on health systems and lessons for future emergency preparedness: a stakeholder analysis in Kenya

Dear Dr. Ogira,

Thank you for submitting your manuscript to PLOS Global Public Health. After careful consideration, we feel that it has merit but does not fully meet PLOS Global Public Health’s publication criteria as it currently stands. Therefore, we invite you to submit a revised version of the manuscript that addresses the points raised during the review process.

We look forward to receiving your revised manuscript.

Kind regards,

Veena Sriram

Academic Editor

Journal Requirements:

Additional Editor Comments (if provided):

Thank you for your patience! The reviewers have noted that the current version of the paper has satisfactorily addressed the feedback raised in the previous round - well done! We are noting that this is a 'minor decision' but the changes requested are very limited. Please note changes suggested by Reviewer 1 in terms of formatting, and also if possible, please see if addressing Reviewer 2's comment about engagement with current literature might be possible (optional). We can turn this around quickly after these minor changes are made.

Reviewers' comments:

Reviewer's Responses to Questions

**Comments to the Author**

1. If the authors have adequately addressed your comments raised in a previous round of review and you feel that this manuscript is now acceptable for publication, you may indicate that here to bypass the “Comments to the Author” section, enter your conflict of interest statement in the “Confidential to Editor” section, and submit your "Accept" recommendation.

Reviewer #1: All comments have been addressed

Reviewer #2: All comments have been addressed

2. Does this manuscript meet PLOS Global Public Health’s publication criteria? Is the manuscript technically sound, and do the data support the conclusions? The manuscript must describe methodologically and ethically rigorous research with conclusions that are appropriately drawn based on the data presented.

Reviewer #1: Yes

Reviewer #2: Yes

3. Has the statistical analysis been performed appropriately and rigorously?

Reviewer #1: N/A

Reviewer #2: N/A

4. Have the authors made all data underlying the findings in their manuscript fully available (please refer to the Data Availability Statement at the start of the manuscript PDF file)?

Reviewer #1: Yes

Reviewer #2: No

5. Is the manuscript presented in an intelligible fashion and written in standard English?

Reviewer #1: Yes

Reviewer #2: Yes

6. Review Comments to the Author

Reviewer #1: The authors have made good efforts to address the review comments. Here are minor comments that can be addressed during the editing stage:

1. Correct the formatting of in-text references 4&5 (Line 59) and references 6&7(Line 61). Both on page 3

2. Correct the grammatical error in the sentence that runs between lines 76 to 79. The sentence is " Although initial mitigation measures aimed at limiting the movement of people through 77 lockdowns and quarantines, in a bid to control infections were put in place. These policy 78 directions also affected the access to other health services such as HIV, tuberculosis (TB), and 79 malaria reversing the gains made in curbing these diseases (19)."

Reviewer #2: Updated draft is much stronger (well done!). The discussion still could benefit from a bit more engagement with wider literature, specifically to contextualize the findings with recent advancements/theory in this area. However, the manuscript still strong and offers a unique contribution to the field.

7. PLOS authors have the option to publish the peer review history of their article (what does this mean?). If published, this will include your full peer review and any attached files.

**Do you want your identity to be public for this peer review?** For information about this choice, including consent withdrawal, please see our Privacy Policy.

Reviewer #1: No

Reviewer #2: No

---

## [Editor Report · Decision Letter 2]

11 Nov 2022

Identifying the impact of COVID-19 on health systems and lessons for future emergency preparedness: a stakeholder analysis in Kenya

PGPH-D-22-00331R2

Dear Ms Ogira,

We are pleased to inform you that your manuscript 'Identifying the impact of COVID-19 on health systems and lessons for future emergency preparedness: a stakeholder analysis in Kenya' has been provisionally accepted for publication in PLOS Global Public Health.

Best regards,

Veena Sriram

Academic Editor

Thank you for your efforts with this manuscript and for your patience!